# Micronuclei-based model system reveals functional consequences of chromothripsis in human cells

**Maja Kneissig[1], Kristina Keuper[1], Mirjam S de Pagter[2], Markus J van Roosmalen[2], Jana Martin[1], Hannah Otto[1], Verena Passerini[3], Aline Campos Sparr[3], Ivo Renkens[2], Fenna Kropveld[2], Anand Vasudevan[4], Jason M Sheltzer[4], Wigard P Kloosterman[2], Zuzana Storchova[1,3]***

[1]Department of Molecular Genetics, TU Kaiserslautern, Kaiserslautern, Germany; [2]Department of Genetics, Center for Molecular Medicine, University Medical Center, Universiteitsweg, Netherlands; [3]Max Planck Institute of Biochemistry, Martinsried, Germany; [4]Cold Spring Harbor Laboratory, Cold Spring Harbor, United States

**Abstract** Cancer cells often harbor chromosomes in abnormal numbers and with aberrant structure. The consequences of these chromosomal aberrations are difficult to study in cancer, and therefore several model systems have been developed in recent years. We show that human cells with extra chromosome engineered via microcell-mediated chromosome transfer often gain massive chromosomal rearrangements. The rearrangements arose by chromosome shattering and rejoining as well as by replication-dependent mechanisms. We show that the isolated micronuclei lack functional lamin B1 and become prone to envelope rupture, which leads to DNA damage and aberrant replication. The presence of functional lamin B1 partly correlates with micronuclei size, suggesting that the proper assembly of nuclear envelope might be sensitive to membrane curvature. The chromosomal rearrangements in trisomic cells provide growth advantage compared to cells without rearrangements. Our model system enables to study mechanisms of massive chromosomal rearrangements of any chromosome and their consequences in human cells.

*For correspondence: storchova@bio.uni-kl.de

Competing interests: The authors declare that no competing interests exist.

## Introduction

Errors in chromosome segregation often result in formation of micronuclei that form to encapsulate chromosomes lagging behind the main chromosome mass. Recent research suggests that micronuclei are not only a consequence of chromosome instability, but further facilitate genomic aberrations (*Hatch and Hetzer, 2015*). This is due to several aspects of micronuclei physiology. First, the nuclear envelope (NE) and nuclear lamina, a dense meshwork of A and B type lamin filaments attached to the inner nuclear membrane, is often aberrant. For example lamin B is lacking in 30–50% of micronuclei, which renders the micronuclei vulnerable to physical forces generated by actin cytoskeleton or during movements of cells through constricted areas, which may result in nuclear rupture (*Hatch et al., 2013*; *Hatch and Hetzer, 2016*; *Irianto et al., 2017*). The compartmentalization disturbed by rupture of NE enables entry of cytoplasmic content that might be harmful for the DNA within the micronuclei (*Hatch et al., 2013*). The NE of micronuclei often lacks other components, such as nuclear pores (*Crasta et al., 2012*; *Zhang et al., 2015*). This impairs the transport of proteins essential for DNA replication or repair, which may hinder stability of the DNA in micronuclei and lead to accumulation of DNA damage (*Terradas et al., 2012*; *Terradas et al., 2009*). Additionally, chromosomes trapped in micronuclei are frequently underreplicated, or their replication occurs later than the replication in the primary nuclei (*Crasta et al., 2012*; *Okamoto et al., 2012*;

*Soto et al., 2018*). These defects may also lead to massive rearrangements in a process called chromoanasynthesis (*Liu et al., 2011*). Chromosomes from micronuclei often rejoin the other chromosomes in the primary nucleus in subsequent cell divisions, which leads to their stabilization within the genome (*Crasta et al., 2012*; *Soto et al., 2018*; *Zhang et al., 2015*).

Chromothripsis, a frequent phenomenon in cancer genomes, enables cancer cells to generate rapid and massive changes in their chromosomal architecture by means of a one-off catastrophic event (*Korbel and Campbell, 2013*; *Stephens et al., 2011*). Chromothripsis breakpoints show blunt fusions or microhomology, indicating repair of fragmented chromosomes through non-homologous or microhomologous repair, suggesting that the rearrangements result from localized chromosome shattering by double-stranded DNA breaks, followed by random reassembly of chromosomal fragments (*Kloosterman et al., 2011*; *Kloosterman et al., 2012*; *Morrison et al., 2014*; *Rausch et al., 2012*; *Stephens et al., 2011*; *Storchová and Kloosterman, 2016*). Chromothripsis has also been found in patients with congenital disorders and even in healthy individuals (*Chiang et al., 2012*; *de Pagter et al., 2015*; *Kloosterman et al., 2012*; *Stephens et al., 2011*). Analysis of patients' genomes showed a different pattern of complex rearrangements involving more than two copy number states, including deletions, duplications and triplications (*de Pagter et al., 2015*). Examination of the breakpoint junctions of these rearrangements revealed microhomology and templated sequence insertions, indicative of the replication-mediated rearrangement mechanism by microhomology-mediated break-induced replication (MMBIR) (*Kloosterman et al., 2014*; *Liu et al., 2011*). Chromosomes with localized rearrangements can also arise during telomere crisis by shattering of dicentric chromosomes. This process often results in a characteristic brakeage-fusion-bridge pattern associated with a burst of localized point mutations (*Maciejowski et al., 2015*). Thus, distinct mechanisms account for chromothripsis.

Micronuclei containing chromosomes were recently pointed out as one of the routes towards chromothripsis-like rearrangements, as chromosomes sequestered to micronuclei with abnormal NE become liable to aberrant DNA repair and replication, which may lead to massive chromosomal rearrangements (*Crasta et al., 2012*; *Ly and Cleveland, 2017*; *Zhang et al., 2015*). Here, we show that microcell-mediated transfer of a chromosome to human acceptor cells often induces a wide range of chromosomal rearrangements, including chromothripsis. Our data show that both replication-based mechanisms and chromosome shattering followed by double-strand break repair occur during chromosome transfer and lead to stably maintained rearrangements. The arising trisomic and tetrasomic cells further propagate despite chromothripsis. Strikingly, we show that chromothripsis provides a growth advantage to trisomic cells compared to cell lines carrying an intact extra chromosome. This previously unappreciated aspect of chromosomal rearrangements may affect selection of cancerous karyotypes.

## Results

### Copy number profiling reveals unique copy number aberrations that specifically affect the transferred chromosome

Microcell mediated chromosome transfer (MMCT) is often used to construct cells with extra chromosomes to analyze the consequences of chromosome gain (*Fournier and Ruddle, 1977*). We created trisomic and tetrasomic cell lines using MMCT in either HCT116 or RPE1 background. The created cell lines were named according to their origin (H: HCT116; R: RPE1), the type of chromosome gain (tr: trisomic; te: tetrasomic) and the transferred chromosome (3, 5, 7, 8, 13, 18 or 21), followed by a sequential number and the information about the presence of H2B-GFP (e.g. Htr3-11_G is a trisomy of chromosome three in the HCT116 with H2B-GFP, clone 11, *Supplementary file 1*). The selection for cells that acquired the extra chromosome was ensured by gene encoding for an antibiotic resistance carried on the extra chromosome. While most cell lines created via MMCT contain an intact extra chromosome, we observed that the integrity of the extra chromosomes was compromised in some cases. To characterize the changes arising from MMCT, we analyzed 38 HCT116-derived and 13 RPE1-derived clonal cell lines carrying extra copies of chromosomes 3, 5, 7, 8, 13, 18 or 21 (*Supplementary file 1*). We determined copy number aberrations (CNAs) > 100 kb on the basis of either Illumina high-density SNP arrays or SMASH whole-genome sequencing (*Figure 1—figure supplements 1* and *2*, *Supplementary file 2*). The supernumerary chromosomes are hereafter referred

to as the *aneuploid chromosomes,* while *transferred chromosome* refers to the specific chromosome copy isolated from the donor cells and transferred via MMCT. In HCT116-derived trisomic and tetra-somic cell lines, we detected 201 CNAs, of which 25 were found also in the parental HCT116 and 176 CNAs were found only in trisomic and tetrasomic cell lines (*Figure 1a*). While 36 of these CNAs were shared among two or more cell lines, we determined a total of 139 unique copy number changes, 58 losses and 81 gains, in 29 of the 38 analyzed aneuploid cell lines. A similar analysis was performed for RPE1- derived cells, where we found only 11 CNAs in both RPE1 parental cell line and its derivatives and 39 CNAs were specific for the RPE1-derived aneuploids (*Figure 1a*). Of these 39 CNAs, 33 were unique for individual trisomies (17 losses and 16 gains) in ten out of 13 RPE1 lines. We noted that the identified unique CNAs were not evenly distributed throughout the genome. Instead, a significant part (41%, 71/172) of these unique CNAs affected only the aneuploid chromo-some, while the disomic chromosomes carried on average only 2.5 CNAs per chromosome (*Supplementary file 2*). Furthermore, CNAs affecting the transferred chromosome were significantly larger than CNAs affecting all other chromosomes (median size 15.8 Mb and 3.3 Mb, respectively, *Figure 1b*).

Since the transferred chromosome is of a different haplotype than the endogenous chromo-somes, we were able to retrieve the affected haplotype from the B allele frequency (BAF) of phase-informative SNPs. We analyzed eight HCT116-derived cell lines that contained up to 25 rearrange-ments (*Supplementary file 3*). In the visualization that depicts the log R ratio and BAF along the chromosome, the SNPs indicating the presence of the transferred chromosome were marked by yel-low dots, while the endogenous SNPs were marked black (*Figure 1c*). Aberrations on the transferred chromosome result in changes of the BAF ratios; for example the absence of yellow dots represents a loss of the transferred chromosome in case of a deletion, whereas the SNPs from endogenous chromosomes (black dots) remain. 97.3% of the CNAs specifically affected the transferred chromo-some in seven clones (*Supplementary file 3*). Only one out of the eight analyzed cell lines (Htr5-07) carries a single unique deletion on one of the endogenous chromosomes. Taken together, these results demonstrate that the transfer of a specific human chromosome in microcells lead to the accu-mulation of unique, *de novo* arising CNAs on the transferred chromosome.

## The rearrangements on the transferred chromosome resemble chromothripsis and chromoanasynthesis

To explore the complexity of the CNAs on transferred chromosomes, we performed whole genome sequencing (WGS) of eight HCT116-derived cell lines: five cell lines that carried multiple *de novo* CNAs on the aneuploid chromosome (Hte5-01, Htr13-03, Htr18-02, Htr21-03 and Htr8-05), and three cell lines with less than two *de novo* CNAs as a control (Htr8-01, Htr8-02, Htr8-07). We called struc-tural variations as well as large copy number changes and analyzed the resulting breakpoint-junc-tions in each of the cell lines. To experimentally validate the breakpoint junctions, we designed primers and successfully obtained a PCR product for 93.6% tested breakpoint junctions (88/94). These breakpoints were further confirmed by Sanger sequencing of the PCR products (*Supplementary file 4*). PCR assays on control samples (A9 and parental HCT116) confirmed that 97.7% of the breakpoint junctions were exclusively present in the respective tri- or tetrasomic clonal cell lines and not in the A9 donors, which indicated that they occurred *de novo*.

Each transferred chromosome was altered by 1 to 33 unique breakpoint junctions (>10 kb), thus indicating a wide range of complexity in genomic rearrangements among the eight analyzed clones (*Figure 1d,e*, *Supplementary file 5*). The number of large rearrangements was substantially higher on the aneuploid chromosome than on any of the other chromosomes in Hte5-01, Htr18-02, Htr21-03 and Htr8-05 (*Figure 1d*). Five of the eight sequenced HCT116 cell lines (Hte5-01, Htr8-05, Htr13-03, Htr18-02 and Htr21-03) showed a complex pattern of rearrangements, comprising multiple breakpoint-junctions and frequent copy number oscillations (*Figure 1e*, *Figure 1—figure supple-ment 3*). This led us to hypothesize that the rearrangements could have resulted from chromothrip-sis (*Korbel and Campbell, 2013*; *Stephens et al., 2011*). Indeed, examination of the breakpoint junctions revealed all four types of orientations (tail-head, head-tail, tail-tail and head-head), as would be expected for chromothripsis (*Figure 1f*).

Moreover, we determined a predominance of microhomologous repair at the breakpoint junc-tions of the complex rearrangement (*Figure 1g*). In the Hte5-01 line, we also observed a high degree of apparently non-templated insertions at the breakpoints, which - together with the

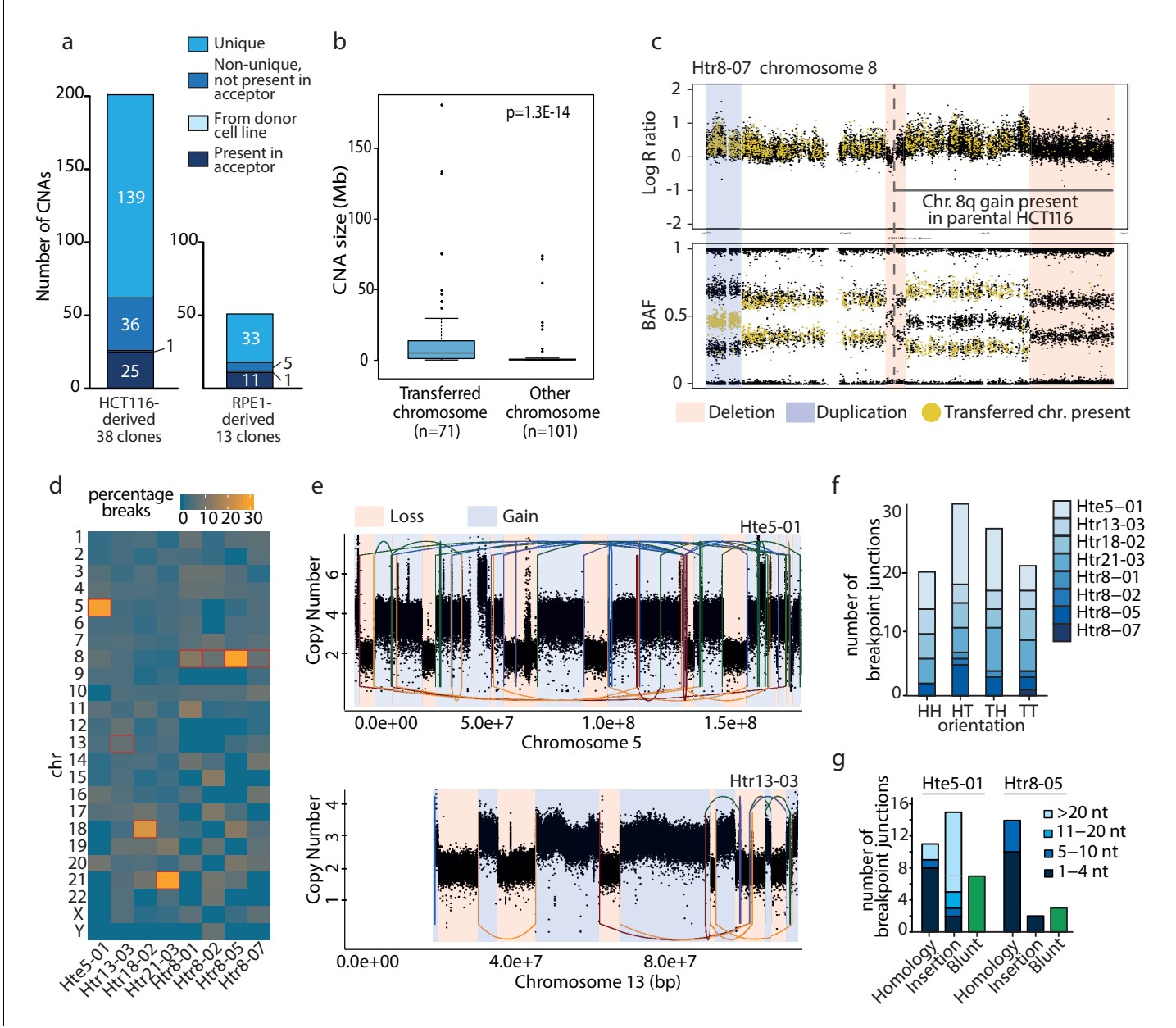

**Figure 1.** Unique *de novo* CNAs specifically affect the transferred chromosome and resemble chromothripsis. (**a**) Detection of CNAs in HCT116- and RPE1-derived cell lines with extra chromosome. A total of 201 (HCT116) and 50 (RPE1) CNAs were found. (**b**) Boxplot showing CNA sizes on the transferred versus the other chromosomes for all cell lines (t-test). (**c**) Deletions (red bars) and duplications (blue bar) on the aneuploid chromosome specifically affect the transferred chromosome. Top and bottom panel: The absence of yellow dots in the deleted regions of the aneuploidy chromosome indicates deletions affecting the transferred chromosome. Bottom panel: the specific change in BAF allows determination of the affected allele for duplications. Here, the shift in BAF from ~0.3 or~0.7 to~0.5 indicates a duplication of the transferred chromosome. (**d**) Heatmap presenting the percentage of breakpoints per chromosome in cell lines analyzed by whole genome sequencing. (**e**) Complex chromosomal rearrangements detected in Hte5-01 (top) and Htr13-03 (bottom). Linear plots show the breakpoint junctions (solid lines) and CNAs detected by WGS. Lines are colored according to the orientation of the breakpoint junction, from low to high chromosomal coordinate: tail-head (blue), head-tail (green), head-head (orange), tail-tail (red). (**f**) Breakpoint junction orientation of the *de novo* structural rearrangements on the aneuploid chromosome. (**g**) Breakpoint characteristics of the breakpoint junctions that were validated to a nucleotide resolution on the aneuploid chromosome in the Hte5-01 and Htr8-05. Colors indicate the amount of nucleotides (nt) for (micro)homology or insertions found at breakpoint junctions.

The online version of this article includes the following figure supplement(s) for figure 1:

**Figure supplement 1.** SNP and SMASH array analysis shows stable addition of the extra chromosome to HCT116 acceptor cells.
**Figure supplement 2.** SNP and SMASH array analysis shows stable addition of the extra chromosome to RPE1 acceptor cells.
**Figure supplement 3.** Copy number and structural rearrangement breakpoint junctions for cell lines analyzed by whole genome sequencing.

occasional blunt fusions - is an indication of break repair by non-homologous or microhomologous end-joining (*Liu et al., 2011*; *Simsek et al., 2011*) and has been noted before for chromothripsis (*Kloosterman et al., 2012*). In addition, the breakpoint junctions' sequences revealed that very short genomic segments were retained in the affected chromosomes following rearrangement formation (*Supplementary file 4*). We detected nine of these short fragments, ranging in size from 63 to 162 bp. Short templated sequences at breakpoint junctions could either be a signature of replicative rearrangement processes (chromoanasynthesis), but they may also occur from shattering and reassembly (*Carvalho et al., 2013*). Taken together, microcell-mediated chromosome transfer can cause complex chromosome rearrangement recapitulating hallmarks of chromothripsis on the transferred chromosome.

## DNA damage occurs in microcells during the transfer

To determine when the DNA damage leading to rearrangements during MMCT occurred, we evaluated each step of the procedure. In MMCT, donor cells (murine cells containing a single specific human chromosome) are treated with microtubule poison colchicine, which drives cells to mitotic slippage during which a single or more chromosomes become captured in micronuclei (MN). These are then isolated as microcells that consist of an intact plasma membrane, a small portion of cytoplasm and a single micronucleus (*Killary and Lott, 1996*). Microcells are subsequently fused to acceptor cells, a colorectal cancer cell line (HCT116) or a telomerase-immortalized human retinal pigment epithelial (RPE1) cell line (*Figure 2a–g*). Incorporation of EdU (5-Ethynyl-2'-deoxyuridine) to DNA encapsulated in the micronuclei confirmed that the chromosome(s) transferred via micronuclei replicated in the acceptor cells. The replication of MN was often asynchronous with the primary nucleus (*Figure 2—figure supplement 1a*), supporting the notion that replication is dysfunctional in MNs. The transferred chromosome was incorporated into the primary nucleus 48 hr – 72 hr after the fusion, which was documented by chromosome painting in the interphase (*Figure 2g*).

We analyzed the micronuclei at each step of the transfer using a single cell gel electrophoresis assay (comet assay), a sensitive technique, in which broken DNA migrates in the electric field towards the anode and creates a characteristic comet profile. We found that the mitotic slippage did not lead to additional DNA damage, as A9 cells treated with colchicine for 48 hr showed similarly low frequency of comets as untreated cells. In contrast, 49.3% of isolated MN showed a comet structure (*Figure 2h,i*). Similar results were obtained using the TUNEL assay that detects DNA fragmentation by labeling the 3-hydroxyl termini. While the donor cells showed only a low level of DNA damage even upon prolonged colchicine and cytochalasin B (DCB) treatment, there was a significant increase of TUNEL-positive isolated MN (*Figure 2j,k*). Moreover, the average intensity of DAPI staining, a proxy for DNA content, was significantly lower in the TUNEL-positive micronuclei than in the TUNEL-negative micronuclei, suggesting either extensive DNA damage or loss through damaged envelope (*Figure 2—figure supplement 1b*). Taken together, the DNA in the isolated microcells becomes frequently damaged.

## Micronuclei lacking lamin B are prone to DNA damage

To determine whether the envelope of the isolated MNs is prone to a rupture, we transfected the donor cells with GFP-tagged cytoplasmic protein (GFP-EB3, cytoplasmic localized signal, CLS) and RFP with nuclear localization signal (RFP-pcDNA4/T0, NLS). These two fluorescent signals are compartmentalized into cytoplasm and nucleus, respectively, but may become mixed upon NE rupture (*Figure 3a*). Evaluation of cells treated with colchicine or colchicine and DCB revealed that the micronuclear envelope is largely intact after mitotic slippage and micronuclei formation (*Figure 3b*). In contrast, we observed colocalization of NLS with CLS in 14% of MNs, while NLS alone was detected only in 2% of the isolated MNs (*Figure 3c*). 84% of the isolated micronuclei were negative for both CLS and NLS, which corresponds with the transfection efficiency. Additionally, the DAPI signal intensity was 3.6 times lower in MNs where both NLS and CLS signals colocalized than in all other MNs (*Figure 2—figure supplement 1c*).

MNs have been shown to often exhibit aberrant localization of lamin B1 proteins, which impairs the structural integrity of the nuclear lamina (*Hatch et al., 2013*; *Okamoto et al., 2012*; *Terradas et al., 2012*). Immunostaining analysis demonstrated that lamin A/C staining was present in nearly all primary nuclei and MN. In contrast, lamin B1 staining was present in all primary nuclei,

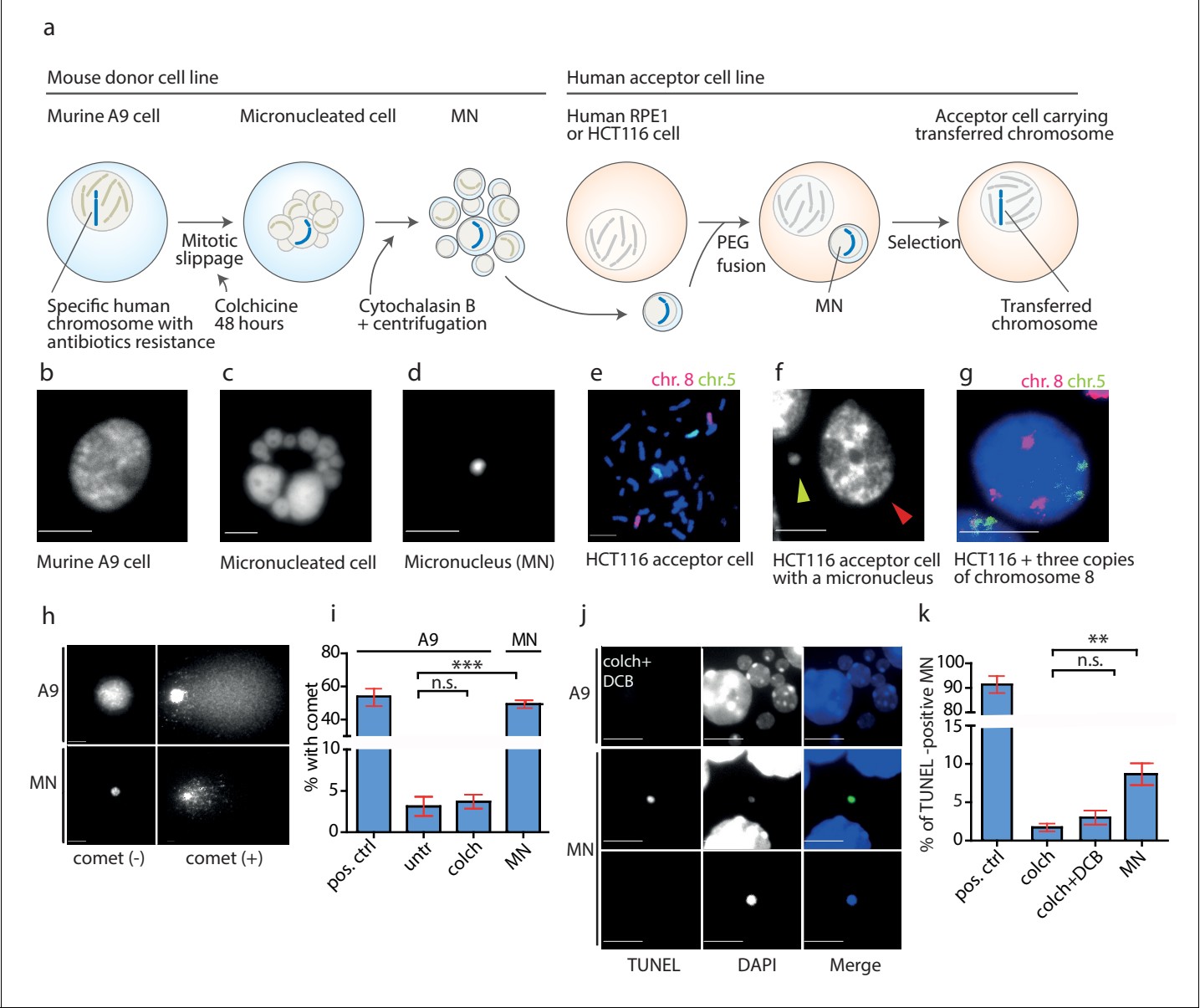

**Figure 2.** DNA damage in isolated micronuclei after mitotic slippage. (a) Schematic depiction of MMCT. (b) Murine A9 cell before micronucleation. (c) Formation of micronuclei in the A9 donor cell upon colchicine treatment. (d) Microcells with micronucleus isolated via centrifugation. (e) Chromosome painting with probes for chromosome 5 (green) and chromosome 8 (red) in HCT116 before fusion. (f) Micronucleus (yellow arrowhead) next to the primary nucleus in the acceptor cell (HCT116, red arrowhead). (g) Chromosome painting of interphase nuclei in trisomic HCT116. (h) Examples of nucleus of an A9 cell and isolated MN without (-) and with (+) DNA apparent comet. DNA was stained with DAPI. (i) Quantification of cells with a comet in untreated cells (untr), cells treated with colchicine for 48 hr (colch) and in isolated MN. Doxorubicin treated A9 cells were used as a positive control (pos. ctrl). All plots show mean ± s.e.m. of three independent experiments; at least 100 nuclei or MN were scored in each experiment. T-test; p<0.0005. (j) Examples of TUNEL assay in A9 cells treated with colchicine for 48 hr followed by 30 min cytochalasin B (colch+DCB) and in isolated MN. DNA was stained with DAPI. (k) Quantification of TUNEL positive MN under conditions as in (c). All plots show mean ± s.e.m. of at least three independent experiments. N: pos. ctrl = 213, colch = 660, colch+DCB = 137, MN = 2003. T-test; p<0.005. Scale bar: 10 μm.

The online version of this article includes the following source data and figure supplement(s) for figure 2:

**Source data 1.** Source data for *Figure 2*.
**Figure supplement 1.** Characteristics of micronuclei.

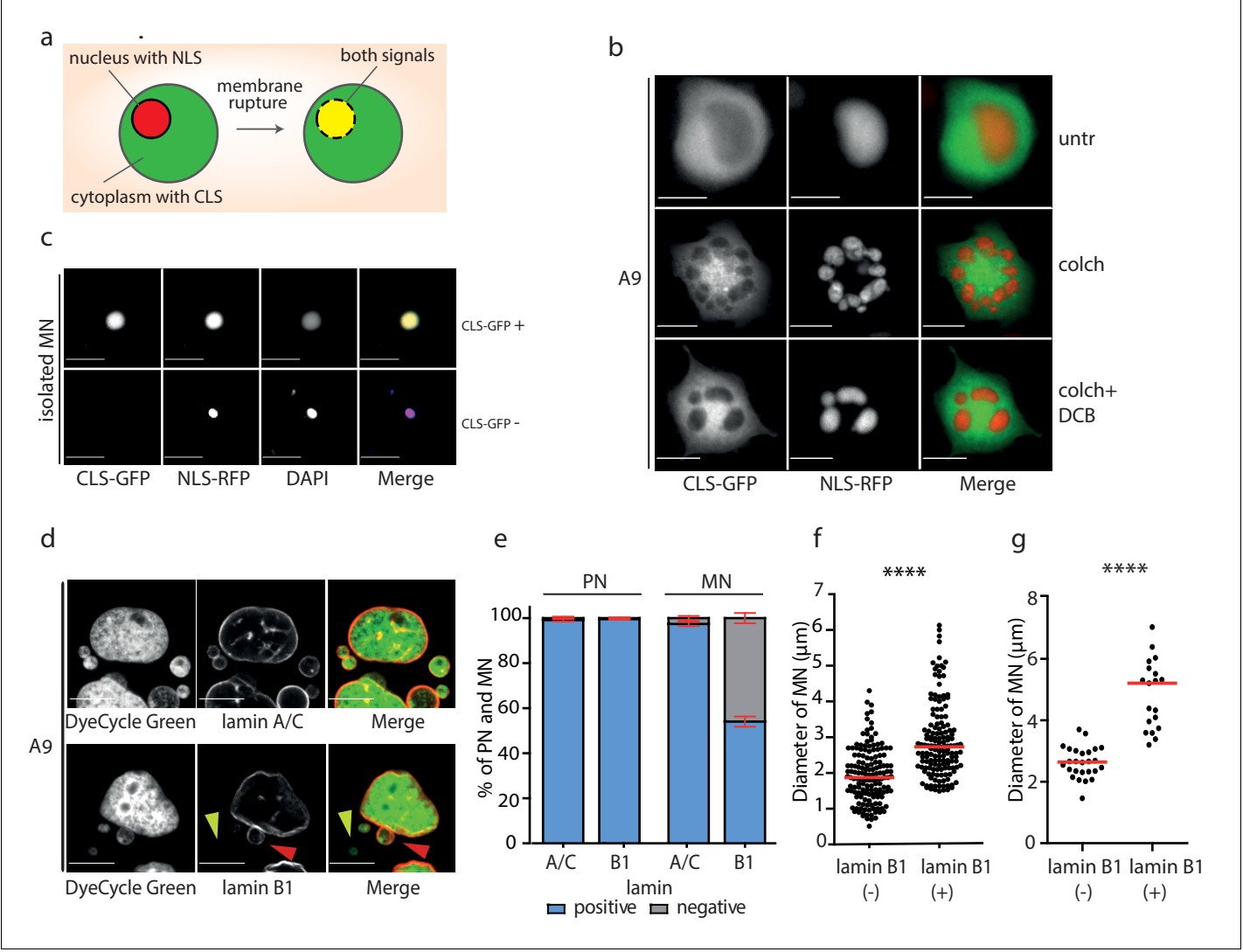

**Figure 3.** Micronuclei frequently lack lamin B1 in the nuclear envelope. (a) Schematic depiction of marker proteins with nuclear localization (red) and cytoplasmic localization (green) after nuclear envelope rupture. (b) Localization of NLS-RFP and CLS-GFP in untreated cells (untr), cells treated with 48 hr of colchicine (colch) and 48 hr of colchicine followed by 30 min of cytochalasin B (colch+DCB) and (c) in isolated MN. (d) Immunofluorescence staining of A9 nuclei and MN with lamin A/C and lamin B1 antibody. The lamin B1 negative MN is highlighted by the yellow arrowhead and the lamin B1 positive MN is indicated by red arrowhead. DNA was stained with DyeCycle Green. (e) Quantification of primary nuclei (PN) and MN positive and negative for lamin A/C and lamin B1. All plots show mean ± s.e.m. of three independent experiments; at least 400 nuclei were scored in each experiment. (f) The diameters of lamin B1 negative (-) and positive (+) MN are visualized. Each dot represents the diameter of one MN. The medians are highlighted in red. Three independent experiments, N = 304. Mann Whitney test, p=0.00012. (g) The diameters of lamin B1 negative (-) and positive (+) MN spontaneously arising in RPE1 cells are plotted. Each dot represents the diameter of one MN. The medians are in red. N = 45. Mann Whitney test, ****p<0.0001. Scale bar: 10 μm in all images.

The online version of this article includes the following source data for figure 3:

**Source data 1.** Source data for *Figure 3*.

but absent in 42% of MN after mitotic slippage (*Figure 3d,e*). The procedure of MN isolation did not affect lamin B1 localization to the NE, since comparable fraction of MN after isolation lacked lamin B1 (49%) (*Figure 2—figure supplement 1d*).

We noticed that the presence of lamin B1 correlates with the size of the micronucleus: while the average diameter of lamin B1 positive micronuclei was 2.7 μm, the average diameter of lamin B1 negative micronuclei was 1.9 μm (*Figure 3f*). The correlation appears to be a general feature, since spontaneously arising micronuclei in RPE1 cells showed a similar trend (*Figure 3g*). The link between

lamin B localization and micronuclei size might suggest that either smaller MNs do not incorporate efficiently lamin B or that the lack of lamin B in the nuclear envelope renders the micronuclei prone to shrinkage. Evaluation of the chromosome number using centromeric antibody revealed that the size of MNs correlates with the number of encapsulated chromosomes. Interestingly, MNs lacking lamin B were consistently smaller than MNs with lamin B regardless the chromosomal content (*Figure 2—figure supplement 1e*). It should be noted that the analysis does not consider the size of the encapsulated chromosome(s) that can vary on an order of magnitude. In conclusion, the nuclear envelope of small MNs usually lacks lamin B. This observation might be explained by our finding that the lack of lamin B renders the MNs prone to further size reduction, possibly due to DNA damage and loss. Additionally, a failure of small MNs to incorporate lamin B efficiently may also contribute to this effect.

The absence of lamin B1 may contribute to the rapture of the NE and accumulation of DNA damage. Indeed, co-staining of MNs with lamin B1 antibody and TUNEL assay showed that DNA damage can be detected in nearly 80% of lamin B1-negative micronuclei, while only 20% of lamin B1-positive micronuclei showed TUNEL signal (*Figure 4a,b*). The occurrence of TUNEL positive micronuclei decreased by 60% when we isolated the microcells at 4˚C. (*Figure 4c*), suggesting that the DNA damage arises through enzymatic activity, most probably of cytoplasmic nucleases. To determine directly the link between NE integrity and the accumulation of DNA damage, we transfected the A9 donor cells with CLS-GFP, induced micronucleation and co-stained the samples with antibodies against lamin B1 and γ-H2A.x and quantified all categories (*Figure 4d*, *Figure 2—figure supplement 1g*). This corroborated that MN lacking lamin B often loose the NE integrity, which was documented by the nuclear localization of CLS (*Figure 4e*). Additionally, DNA damage was detected in nearly 50% of lamin B1-negative micronuclei, while only 22.5% of lamin B1-positive micronuclei showed γ-H2A.x signal (*Figure 4f*). Finally, DNA damage (γ-H2A.x signal) was identified in 58% of MNs with nuclear localization of CLS, while less than 20% of MNs without nuclear CLS showed DNA damage (*Figure 4g*).

Finally, we asked whether the mechanical stress during MN isolation (centrifugation, filtration) contributes to the NE rupture and subsequently to the accumulation of DNA damage. To this end, we compared the occurrence of DNA damage in isolated MNs with the DNA damage in MNs that were incubated in medium with DCB for equal time. These data showed an increased accumulation of DNA damage with incubation time as determined by γ-H2A.x signal. Importantly, the MN from samples that underwent centrifugation steps showed elevated γ-H2A.x signal (*Figure 2—figure supplement 1f*). Taken together, we propose that micronuclei that arise from mitotic slippage in the absence of microtubules fail to properly localize lamin B1 to the NE. This renders the envelope sensitive to mechanical stress during MN isolation and increases the probability of nuclear rupture and additional DNA damage.

## Functional consequences of massive chromosomal rearrangements

The cognate aneuploid cell lines with and without chromothripsis allowed us to determine the functional consequences of chromosomal rearrangements. Gain of a chromosome generally impairs cellular proliferation in human cells and most trisomic human cells proliferate slower than their isogenic parental cell line (*Sheltzer et al., 2017*; *Stingele et al., 2012*). Strikingly, we found that the cell lines with a rearranged chromosome formed colonies in soft agar more readily than the cell lines with an identical, but intact chromosome (*Figure 5a,b*). We considered several possibilities to explain this observation. First, chromosomal rearrangements can result in a loss of anti-proliferative genes or a gain of pro-proliferative factors. Using the TUSON database (*Davoli et al., 2013*), we mapped all previously defined tumor suppressor genes (TSGs) and oncogenes (OGs) on the extra chromosomes. As most rearrangements results in a loss of genomic material compared to fully trisomic cell lines, we hypothesized to observe more OGs and less TSGs overlapping with the chromosome gain regions. Analysis of chromosome 5, 13 and 18 in trisomic cell lines with and without rearrangements did not suggest any bias towards gain of OGs and loss of TSGs (*Figure 5—figure supplement 1a*). However, an effect of specific rearrangements cannot be excluded. For example, individual assessment of the analyzed cell lines revealed that Htr13-03 (with rearranged chromosomes 13) lost the extra copies of *COL4A2* and *COL4A1*, while Htr13-02 (with intact chromosome 13) carries three copies of each (*Figure 5—figure supplement 1b*). *COL4A2* and *COL4A1* encode type IV collagen alpha proteins. Upon proteolytic cleavage of COL4A2 and COL4A1, biologically active peptides called

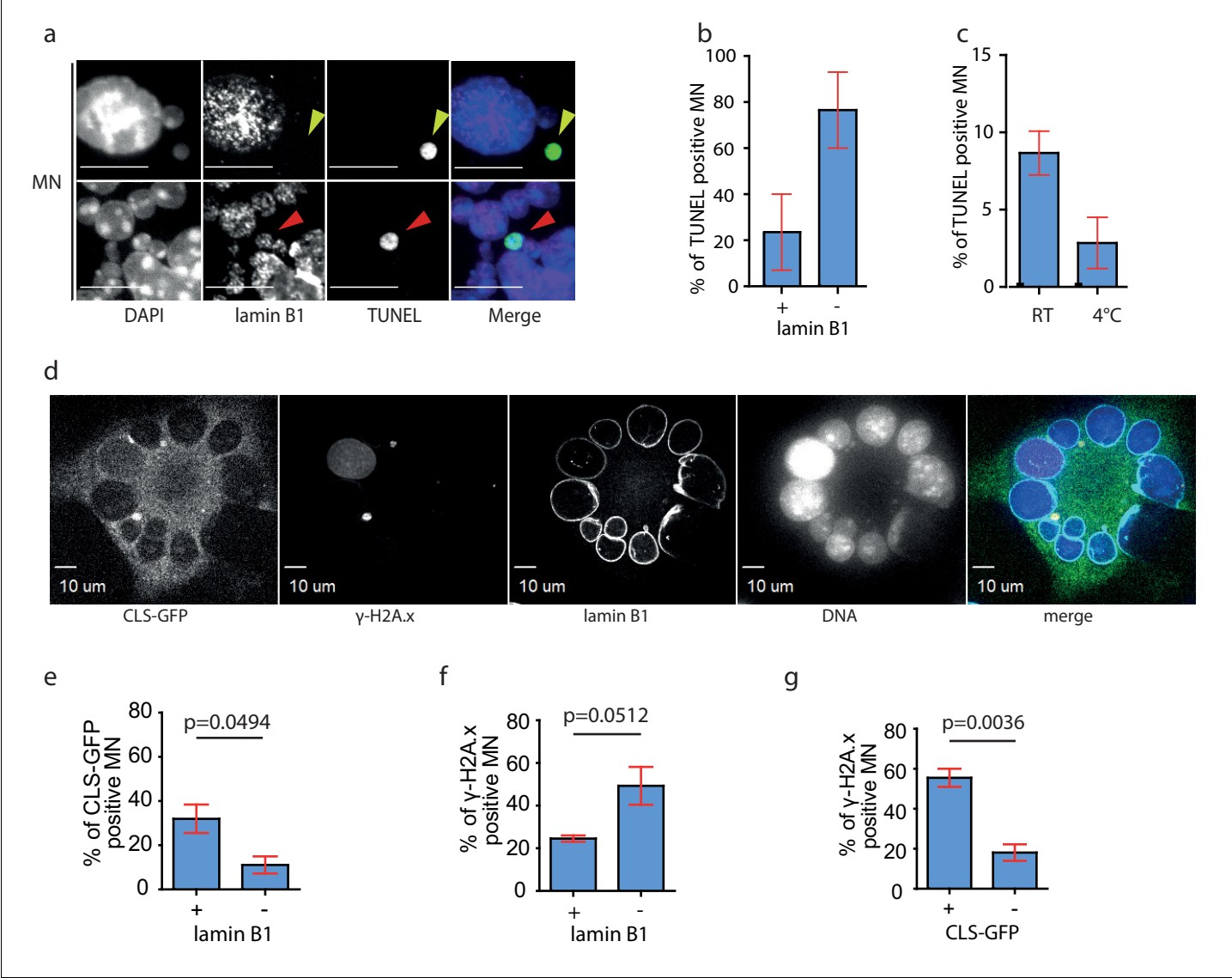

**Figure 4.** DNA damage occurs predominantly in lamin B1 negative MN. (**a**) Immunofluorescence staining of isolated MN with lamin B1 antibody combined with TUNEL assay. The lamin B1 negative MN is highlighted by yellow arrowhead, the lamin B1 positive MN is indicated by red arrowhead. DNA was stained with DAPI. Scale bar: 10 µm. (**b**) Percentage of TUNEL in lamin B1 positive (+) and lamin B1 negative (-) MN. Plots show mean ± s.e.m. of two independent experiments. N = 58. (**c**) Distribution of TUNEL positive MNs after their isolation at RT and at 4°C. Plots show mean ± s.e.m. of three experiments in RT and two at 4°C. Total N: RT = 1292; 4°C = 1918. (**d**) Representative image of micronucleated A9 cell after transfection with CLS-GFP plasmid followed by colchicine treatment and immunouorescence staining of –H2A.x and lamin B1. DNA was stained with DAPI. Scale Bar: 10 µm. (**e**) Percentage of CLS-GFP positive MN in lamin B1 positive (+) or negative (-) MN. (**f**) Percentage of –H2A.x positive MN in lamin B1 positive (+) and negative (-) MN. (**g**) Percentage of –H2A.x positive MN in CLS-GFP positive (+) and negative (-) MN. (**e**)-(**g**): Unpaired t-test was performed, p-values are shown (p). Total N: 295 MN. Plots show mean ± s.e.m. of three independent experiments.

The online version of this article includes the following source data for figure 4:

**Source data 1.** Source data for *Figure 4*.

Arresten and Canstatin, respectively, are formed that inhibit angiogenesis, proliferation and tumor formation (*Egeblad et al., 2010*; *Kamphaus et al., 2000*). Further experiments will be necessary to determine whether copy number changes or rearrangements of individual genes could be selected to improve proliferation of aneuploid cells.

Second, we considered a possibility that the massive chromosomal rearrangements may further destabilize the genome, which in turn facilitates adaptation to trisomy. Therefore, we analysed the occurrence of mitotic errors by scoring the spontaneous formation of MNs and anaphase bridges.

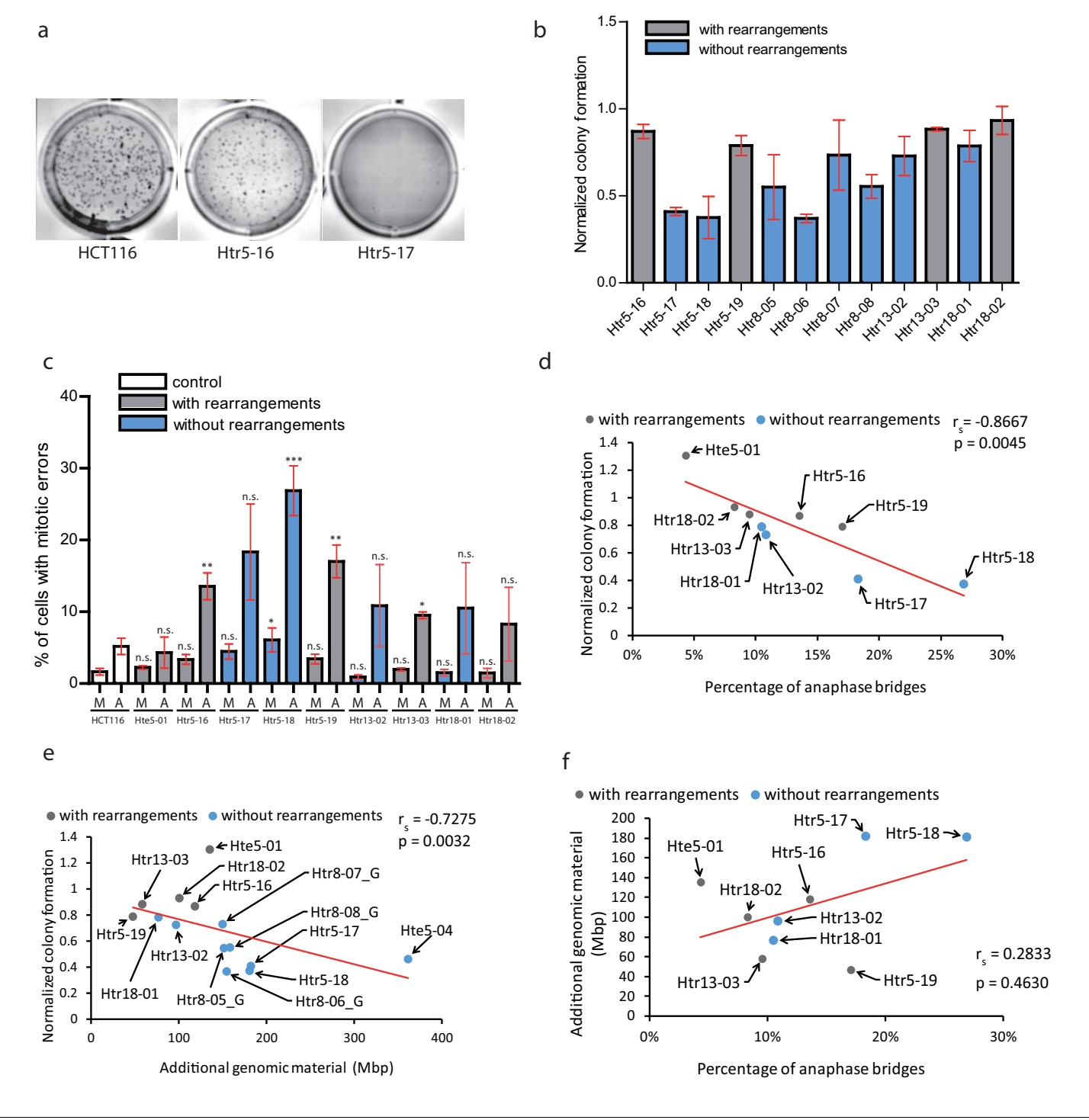

**Figure 5.** Chromothripsis provides functional advantages to the cells. (**a**) Example of soft agar assay to determine colony formation ability in parental cell line HCT116, trisomy with a highly rearranged extra chromosome 5 (Htr5-16) and trisomy with a nearly intact extra chromosome 5 (Htr5-17). (**b**) Number of colonies in trisomic cell lines with (gray) and without (blue) rearrangements on the extra chromosome, normalized to the parental control. Each bar represents mean ± s.e.m. of four replicates. (**c**) Quantification of micronuclei (**M**) and anaphase bridges (**A**) in trisomic cell lines with (gray) and without (blue) rearrangements on the extra chromosome. Each bar represents mean ± s.e.m. of at least three independent experiments. (**d**) The normalized colony forming capacity of the cell lines correlates with the percentage of anaphase bridges in trisomic and tetrasomic samples. (**e**) The amount of additional genomic material in trisomic and tetrasomic samples correlates with the normalized colony forming capacity of the cell lines. (**f**) The amount of additional genomic material in trisomic and tetrasomic samples correlates with the percentage of anaphase bridges. The cell lines with

*Figure 5 continued on next page*

*Figure 5 continued*

rearrangements on the extra chromosome are marked in grey, the cell lines without rearrangements are in blue. Data in (**c**) were evaluated using unpaired T-test (compared to HCT116). P-value: *<=0.0154 **<=0.0017 ***<=0.0004. Data in (**d**) - (**f**) were evaluated using Spearman Rank Correlation. Correlation coefficient ($r_s$) and two-tailed p-values (p) were calculated. Differences of the datasets in (**d**) and (**e**) are significant.

The online version of this article includes the following source data and figure supplement(s) for figure 5:

**Source data 1.** Source data for *Figure 5*.
**Figure supplement 1.** Gene-specific changes.
**Figure supplement 2.** Correlation of colony forming capacity.

As expected, trisomic and tetrasomic cells display more mitotic errors, in particular increased frequency of anaphase bridges compared to parental cell lines (*Passerini et al., 2016*) (*Figure 5c*). Strikingly, the accumulation of anaphase bridges and micronuclei negatively correlates with the colony formation, suggesting that increased genomic instability impairs cellular proliferation in soft agar (*Figure 5d*). Previously, it has been shown that the phenotypes of trisomic cells are impaired due to the expression of extra proteins and therefore correlate with the amount of extra DNA (reviewed in *Chunduri and Storchová, 2019*). We therefore correlated the amount of extra DNA with the ability to form colonies on soft agar, demonstrating that the less extra DNA the cells contain, the more colonies they form (*Figure 5e*). Interestingly, the occurrence of anaphase bridges and chromosome missegregation also correlated with the amount of additional genomic material (*Figure 5f*, *Figure 5—figure supplement 2b*). This analysis suggests that the proliferation of trisomic cell lines is negatively affected by genomic instability. Additionally, it reveals a previously unappreciated aspect of massive chromosomal rearrangements, which is the elimination of extra DNA that would otherwise present a burden for human cells.

## Discussion

Genomic rearrangements are a major hallmark of cancer, yet their origin and molecular causes are often unclear. Here, we present a method that generates cell lines with individual chromosomes altered by genomic rearrangements through capture of a human chromosome in a micronucleus. We show that chromosomes trapped within micronuclei can undergo a wide range of genomic rearrangements, from a few simple rearrangements, to complex chromosomal reshuffling resembling chromothripsis (*Figure 1*). This was particularly prominent in HCT116-derived trisomic cell lines, where 66% of them showed more than one rearrangement on the transferred chromosome. Trisomic cell lines derived from non-transformed RPE1 carry markedly less *de novo* rearrangements (compare *Figure 1—figure supplements 1* and *2*). This might be due to the fact that the non-transformed cells with fully functional checkpoints are less tolerant to DNA damage and aneuploidy than cancerous cells. Importantly, these rearrangements were not identified in the donor cell lines before the transfer, thus supporting the idea that the passage through micronuclei contributed to their formation. It should be noted that the donor cell lines (A9 murine cell lines) were also created by chromosome transfer and therefore might also carry previous rearrangements. This however, is excluded by two facts: (1) Only one common CNA was identified on the transferred chromosome, suggesting its origin in the common donor (*Figure 1a*). (2) PCR over the breakpoint junctions did not reveal their presence in the donor cell lines (*Supplementary file 4*). On the basis of the genomic hallmarks of the induced rearrangements, we propose that the rearrangement formation within micronuclei during MMCT may result from aberrant replicative processes (serial template switching) as well as from chromosome shattering by double-strand breaks, followed by DNA repair (*Kloosterman et al., 2012*; *Stephens et al., 2011*; *Holland and Cleveland, 2012*; *Liu et al., 2011*). Our data are in line with the recent findings that revealed the role of micronuclei as a source of genomic instability (*Crasta et al., 2012*; *Soto et al., 2018*; *Zhang et al., 2015*). Thus, we demonstrate that MMCT may serve as a model system for analysis of mechanisms involved in chromothripsis of specific chromosomes and their consequences. Although MMCT is an artificial system to generate micronuclei, we note that this is to our knowledge the only model system that allows targeted generation of chromothripsis of any chromosome in human cell lines.

Secondly, we show that the chromosomal rearrangements occurred on the specific chromosome entrapped in the micronuclei and used for the MMCT. Nearly 50% of the micronuclei isolated for

MMCT lacked lamin B1, which promotes rupture of the micronucleus and entry of the cytosol components into the micronucleus (*Figure 3d,e*). A significant proportion of MN that arise due to chromosome missegregation around lagging chromosome rupture during interphase in response to mechanical forces (*Hatch et al., 2013*). We propose that the mechanical forces that arise during the isolation procedure may also facilitate rupture of the MNs. This in turn leads to defects in DNA metabolism, loss of compartmentalization, entrance of cytoplasmic nucleases and massive DNA damage. Additionally, DNA damage or replication-dependent rearrangements can occur in MN after fusion to the acceptor cells, as suggested by the finding of patterns of aberrant replication processes on the rarranged chromosomes in trisomic cells (*Figure 1*).

Why is the nuclear lamina dysfunctional in micronuclei? Previous works suggested that Aurora B kinase activity affects assembly of the NE around lagging chromosomes and interferes with efficient lamin B deposition via so called 'chromosome separation checkpoint' (*Afonso et al., 2014*). Alternatively, microtubuli of the mitotic spindle interfere with the proper assembly of NE around lagging chromosomes in late stages of mitosis (*Liu et al., 2018*). Our data are not consistent with these models. First, the micronuclei created in MMCT form during mitotic slippage due to a treatment with microtubule poison, which reduces the probability of the interference of microtubules with NE assembly. Second, the micronuclei arise after nuclear slippage, where no chromosome separation took place. Instead, we observed that micronuclei with a small diameter lack lamin B1 more often than larger micronuclei (*Figure 3f,g*). This suggests that multiple processes affect formation of functional nuclear envelope. We hypothesize that a higher curvature of the NE in MN may negatively affect the lamina assembly, similarly as it was demonstrated in primary nuclei where high membrane curvature impairs nuclear lamina (*Denais et al., 2016*).

Finally, the micronuclei-mediated rearrangements obtained via MMCT can be stably propagated in cell lines, allowing the analysis of their consequences. A gain of a chromosome is often detrimental for human cells, which is reflected by impaired proliferation (*Sheltzer et al., 2017*; *Stingele et al., 2012*). Comparison of proliferation of cell lines carrying intact extra chromosomes with cell lines carrying rearranged chromosomes revealed that the chromosomal rearrangements reduce the negative effects of chromosome gain on cellular proliferation by reducing the amount of extra DNA (*Figure 5*). Previous work suggested that chromothripsis improves cellular proliferation in soft agar, however, it is important to note that the experiments were performed in tetraploid background (*Mardin et al., 2015*) and tetraploidy alone facilitates proliferation in soft agar (*Fujiwara et al., 2005*; *Kuznetsova et al., 2015*). Here we demonstrate for the first time that chromothripsis provides growth advantage also in near-diploid cells. This is on one hand due to a previously unappreciated effect of chromosomal rearrangements, which is a removal of unnecessary genetic material, and we propose that this aspect may be another important factor that shapes cancer genomes. At the same time, our data reveal that increased genomic instability is an important factor that negatively affects proliferation of trisomic cells. Thus, a vital question of future experiments will be how aneuploid cells adapt to elevated genomic instability.

## Materials and methods

### Cell culture and treatment

RPE-1 hTERT (referred to as RPE-1) and RPE-1 hTERT H2B-GFP were a kind gift of Stephen Taylor (University of Manchester, UK). HCT116 H2B-GFP was generated by lipofection (FugeneHD, Roche) of HCT116 (ATCC No. CCL-247) with pBOS-H2B-GFP (BD Pharmingen) according to manufacturer´s protocols. Trisomic and tetrasomic cell lines were generated by microcell-mediated chromosome transfer as described below. The cell line Hte5_01 was kindly provided by Minoru Koi, Baylor University Medical Centre, Dallas, TX, USA. The A9 donor mouse cell lines were purchased from the Health Science Research Resources Bank (HSRRB), Osaka 590–0535, Japan. All cell lines were maintained at 37° C with 5% $CO_2$ atmosphere in Dulbecco´s Modified Eagle Medium (DMEM) containing 10% fetal bovine serum (FBS), 100 U penicillin and 100 U streptomycin. All cell lines were determined negative for mycoplasma contamination.

## Microcell-mediated chromosome transfer

A9 mouse cells containing human chromosomes of various origin (chromosome 3, 7 – NTI-4, human fetus-derived fibroblast cell line; chromosome 5, 8, 18 - Mo, human T- lymphocyte cell line; Chromosome 13, 21 – unspecified) and marked with a gene for antibiotic resistance were used as donors. Briefly, the cells were treated for 48 hr with 50 ng/ml colchicine at 37°C. The cells were then collected, evenly distributed on plastic slides and incubated for 3 hr at 37°C. Next, the slides were centrifuged for 30 min in pre-warmed DMEM containing 10 µg/ml DCB. The pellet was re-suspended in DMEM and filtered twice with 8 and 5 µm filters. The isolated micronuclei were fused with the recipient cell lines. Cells with the extra chromosome were selected in DMEM with the appropriate antibiotic. For detailed description see *Stingele et al. (2012)*. All trisomic and tetrasomic cell lines originate from a single cell. The arising colonies were collected and expanded for four to six weeks and aliquots were frozen for each cell line. For each experiment, a new aliquot was thawed and used for a limited number of passages (less than five). Obtained cell lines were named according to their parental cell line (H: HCT116; R: RPE1), the tri- or tetrasomic state of the aneuploid chromosome (tr: trisomic; te: tetrasomic) and the chromosome that was transferred (3, 5, 7, 8, 13, 18 or 21), followed by a sequential number and the information about the presence of H2B-GFP (e.g. Htr3-11_G for trisomy of chromosome 3 in the HCT116 with H2B-GFP and clone number 11, *Supplementary file 1*).

## Metaphase spread for karyotyping

Cells were grown to 70–80% confluency, treated with 50 ng/ml colchicine for 3–5 hr, collected by trypsinization and centrifuged at 1000 rpm for 10 min. Pellets were resuspended in 75 mM KCl and incubated for 10–15 min at 37°C. After centrifugation, the pellets were resuspended in 3:1 methanol/acetic acid to fix the cells, washed several times in 3:1 methanol/acetic acid, spread on a wet glass slide and air dried at 42°C for 5 min. Each sample was labeled with probes for two different chromosomes, a transferred chromosome and a control chromosome, according to the manufacturer's instructions (Chrombios GmbH, Raubling, Germany) and counterstained with DAPI.

## Analysis of DNA synthesis by EdU incorporation

EdU (10 µM) was added to HCT116 and RPE1 cells at 20 hr after fusion with isolated micronuclei. For EdU detection, cells were fixed for 15 min with 3.7% paraformaldehyde. Next, cells were permeabilized for 15 min with 0.1% Triton X-100 in PBS and incubated with EdU Click-iT reaction mix (Invitrogen) according to the manufacturer's instructions. The DNA was labelled with 1.0 µg/ml DAPI or SytoxGreen (2.0 µM) and cells were imaged with the fluorescence microscope mentioned below.

## Single cell gel electrophoresis assay (comet assay)

The alkaline comet assay was performed as described previously (*Tice et al., 1990*). Microscopic slides were covered with 1% normal agarose and left to solidify. The first agarose layer was covered with cell suspension of $1 \times 10^4$ cells in 10 µl PBS mixed with 75 µl of 1% low melting point agarose at 37°C and immediately covered with the coverslip. Slides were placed on ice to solidify the agarose. Afterwards, the third layer of 0.5% low melting point agarose at 42°C was added and covered with the coverslip. After solidification on ice, the slides were immersed in the lysing solution (2.5 M NaCl, 100 mM EDTA, 10 mM TRIS, pH 10) for 2 hr at 4°C. The slides were then transferred to an electrophoresis solution (300 mM NaOH, 1 mM EDTA) for 20 min. Next, electrophoresis was carried out at 21 V, 150 mA for 20 min. The slides were then washed in neutralization solution (0.4 M Tris, pH 7.5) three times for 5 min and dehydrated in 70%, 90% and 100% ethanol, 3 min each. The slides were stained with 1.0 µg/ml DAPI and imaged with a fluorescence microscope.

## TUNEL assay

To perform TUNEL assay, isolated MN were fixed in ice-cold methanol and acetic acid (3:1) three times for 1 min. Fixed MN were dropped on a wet glass slide and air dried at RT. TUNEL assay was performed according to the In Situ Cell Death Detection Kit (Roche Diagnostics) instruction. Briefly, MN were incubated in permeabilization solution for 2 min on ice. For positive control, fixed and permeabilized MN were incubated with 1000 Units/ml DNase I recombinant for 10 min at RT. The TUNEL reaction mixture was added on the slide, covered with parafilm and incubated in a

humidified atmosphere for 1 hr at 37°C. MN were counterstained with 1.0 µg/ml DAPI and imaged with a fluorescent microscope or used for further immunofluorescence staining.

## Immunofluorescence staining

The cells were seeded in flat, black 96-well glass-bottom plates and grown in DMEM to the desired confluence of 70–80%; fixed with freshly prepared 3.7% formaldehyde for 12 min at RT and permeabilized with 0.5% Triton X-100 in PBS for 5 min. Next, the cells were blocked in 10% FBS and 0.1% Triton X-100 for 30 min at RT and co-immunostained with ant-lamin A/C (1:50; Santa Cruz sc-376248) and anti-lamin B1 (1:500; Abcam ab16048) overnight at 4°C, followed by a secondary anti-mouse antibody (Alexa Fluor 594 1:1000; Jackson ImmunoResearch 715-858-150) and anti-rabbit antibody (Alexa Fluor 647 1:1000; Jackson ImmunoResearch 711-605-152) for 1 hr at RT. Alternatively, the following antibody combinations were used: To visualize Lamin B and γ-H2A.x the A9 samples were co-immunostained with anti-γ-H2A.x (1:10000; Abcam ab26350) and anti-lamin B1 (1:500; Abcam ab16048) overnight at 4°C, followed by a secondary anti-mouse antibody (Alexa Fluor 594 1:1000; Jackson ImmunoResearch 715-858-150) and anti-rabbit antibody (Alexa Fluor 647 1:1000; Jackson ImmunoResearch 711-605-152) for 1 hr at RT. To visualize Lamin B and the centromeres the A9 samples were co-immunostained with anti-centromere (1:500; ImmunoVision HCT0100) and anti-lamin B1 (1:500; Abcam ab16048) overnight at 4°C, followed by a secondary anti-human antibody (Alexa Fluor 647 1:1000; Jackson ImmunoResearch 715-858-150) and anti-rabbit antibody (DyLight 405 1:1000; Jackson ImmunoResearch 711-475-152) for 1 hr at RT. The DNA was counterstained with DyeCycle Green (1:1000), SytoxGreen (2.0 µM) or 1.0 µg/ml DAPI and imaged as described below. For MN analysis, isolated MN were fixed in ice-cold methanol and acetic acid (3:1) three times for 1 min and dropped on a wet glass slide. The subsequent protocol was identical as above.

## Lipofectamine transfection

Lipofectamine 2000 (Thermo Fischer Scientific) transfection was performed according to the manufacturer's instruction. Briefly, cells were seeded in a 6-well or 96-well plate to 70–90% confluency on the day of transfection. The reagent was mixed with Opti-MEM Medium and 1 µg of DNA (or 2 µg in case of the 6-well plate) was diluted in Opti-MEM Medium and incubated for 5 min at RT. Diluted Lipofectamine 2000 reagent was added to the diluted DNA (1:1 ratio) and incubated for 15 min at RT. Finally, DNA – lipid complex was added to the cells in the form of droplets. The cells were then incubated overnight at 37°C and used for further experiments.

## Soft agar colony forming assay

1% low melting agarose combined with an equal volume of DMEM was added to 3.5 cm dishes and solidified for the bottom layer. 0.7% low melting agarose was mixed with an equal volume of cell suspension containing 1000 cells/ml and immediately spread on the bottom agar. Following solidification at room temperature, culture was propagated for 14 days in a 37 °C incubator with 5% $CO_2$. The dish was then divided into four fields and colonies in each field were counted using inverted microscope (Motic AE2000).

## Microscopy

Images were obtained by a fully automated Zeiss inverted microscope (AxioObserver Z1) equipped the CSU-X1 spinning disk confocal head (Yokogawa) and LaserStack Launch with selectable laser lines (Intelligent Imaging Innovations, Denver, CO). Image acquisition was performed using a Cool-Snap HQ camera (Roper Scientific) and a 40x air or 63x oil objective under the control of the Slide-book6 software (Intelligent Imaging Innovations, Denver, CO).

## DNA isolation

RPE1 and HCT116 cells and their derivatives were harvested by trypsinization and centrifugation. Cells were washed two times with PBS and DNA was subsequently isolated using the Qiagen DNA easy or Qiagen QIAamp kit (Cat. No. 51036) following manufacturer's protocol.

## SNP array and SMASH analysis

For each sample, 200 ng DNA was used as input for copy number profiling using Cyto12 SNP arrays according to standard procedures (Illumina). Copy number aberrations were called using Nexus Copy Number 6.0 software (BioDiscovery) followed by manual curation. SNP array log ratios of signal intensities, allele frequencies and genotypes were extracted from Illumina main project files using Genome Studio software (Illumina). Coordinates are in hg19. Copy number variations from SNP array data were predicted using Penn CNV (version 2014) (*Wang et al., 2007*).

SMASH (Short Multiply Aggregated Sequence Homologies) karyotyping was performed as previously described (*Wang et al., 2016*). Briefly, total genomic DNA was enzymatically fragmented to an average size of ~40 bp and ligated to create chimeric fragments of DNA suitable for creating NGS libraries (300–700 bp). The fragment size selection was performed with Agencourt AMPure XP beads (Beckman Coulter, Cat. No. A63881). Illumina-compatible NEBNext Multiplex Dual Index Primer Pairs and adapters (New England Biolabs, Cat. No. E6440S) were ligated to the selected chimeric DNA fragments. The barcoded DNA fragments were sequenced on an Illumina MiSeq. The generated reads were demultiplexed and mapped using custom scripts, and plots were generated with G Graph (*Wang et al., 2016*). Calling of CNAs within the SMASH data was done by assigning gains or losses based on the SMASH seq-quantal copy number value. Segments with a seq_quantal value <1.5 were regarded as losses and segments with a value >2.5 were regarded as gains. For segments on the aneuploidy chromosome, the seq_quantal value was corrected by subtracting 1. Segments were merged based on similar seq_quantal values, resulting in copy number calls. SNP array and SMASH CNA segments were further merged two, or more consecutive segments cover more than 90% over their total genomic span. Unique CNAs were subsequently filtered out and were defined as having less than 70% reciprocal overlap with any other CNA detected in a clone originating from the same parental cell line or the corresponding parental cell line itself (RPE1 or HCT116).

## Determination of the affected allele for CNAs

We used phase-informative SNPs from SNP array data to determine which allele was affected by CNAs as a result of chromosome transfer. First, we compared the SNP allele frequencies from trisomic cells with corresponding parental HCT116 cells to detect SNP positions that are homozygous ($A^n$ or $B^n$, Illumina B allele frequency <0.05 or>0.95) in the cells (Illumina B allele frequency of 0 or 1, AA or BB genotype) and have a different SNP allele on the transferred chromosome (change of B allele frequency to 0.33 or 0.66, AAB or ABB genotype). Subsequently, we examined genomic intervals with unique copy number changes on the chromosome used for transfer and determined the SNP allele frequencies within these intervals. In case of a deletion on the transferred homologue, the genotype would change from AAB/ABB in normal trisomic cells (without chromosomal rearrangements on the transferred chromosome) to AA or BB in the rearranged trisomic cells. If a deletion had occurred on one of the two endogenous chromosomes present in the parental HCT116 cells, an AB genotype would be expected. Similarly, in case of a duplication, the AAB or ABB genotype in the normal trisomic cells will shift to AABB if the duplication occurs on the transferred homologue and to AAAB or ABBB if the duplication occurs on one of the two homologous chromosomes present in the parental HCT116 cells. We set the threshold for the shift to a BAF >0.15 and<0.85 SNPs that show this shift are indicated by a yellow dot in Log R ratio and B allele frequency plots.

## Whole-genome sequencing and breakpoint detection

Whole-genome sequencing was performed at the Garvan Institute of Medical Research and Novogene. For the Garvan sample processing, 2 µg of genomic DNA was used for library preparation and paired-end sequenced (2 × 150 bp) to an average depth of >30 x on an Illumina HiSeq X Ten platform. For samples Htr13-03, Htr18-02 and Htr21-03, DNA was sequenced on Illumina NovaSeq. Sequence reads were mapped using BWA with settings bwa mem –c 100 –M –R. Subsequently, duplicated reads in BAM files were marked using Sambamba (*Tarasov et al., 2015*). Deduplicated BAM files were used as input for detection of genomic rearrangements using Manta (*Chen et al., 2016*), with default settings.

## Rearrangement validation by PCR and Sanger sequencing

Primers for validation of breakpoint junctions were designed using primer3 software. All primers were filtered against a reference set of human repetitive elements to improve specificity of the PCR reaction. PCR was performed using Phusion polymerase (New England Biolabs) or Amplitaq Gold polymerase (Life Technologies). We used 5–10 ng native genomic DNA for each PCR reaction and used a touchdown PCR with an annealing temperature decreasing from 65°C to 58°C (0.5°C per cycle). PCR products were subjected to Sanger sequencing to confirm breakpoint authenticity and to classify breakpoint sequence characteristics.

## Breakpoint fine mapping using local *de novo* assembly

For the mapping, we used *de novo* assembly of discordant read pairs to assemble contigs that span the rearrangement breakpoints. To perform local *de novo* assembly of breakpoint regions, we first used sambamba (*Tarasov et al., 2015*) to extract discordant read pairs from genomic regions that were identified by discordant pair analysis based on the following commandline: sambamba view -f bam -L < bedfile > F 'not proper_pair' -o < output > <inputbam > . Next, we used SOAPdenovo (*Luo et al., 2012*) to assemble the discordant reads from a bam file containing all the extracted discordant reads. We tested different k-mer sizes for the assembly process to enable assembly of as many breakpoints as possible. The contigs that resulted from the assembly were blasted to the GRCh37 reference genome using NCBI megablast (version 2.2.31+). To determine breakpoint features, blast output of assembled breakpoint sequences was manually aligned to sequences of the reference genome on either side of the breakpoint junction.

## Determination of the affected allele from WGS reads

We selected phase-informative SNPs from the genome sequencing data and subsequent Sanger sequencing of SNP positions in the A9 donor lines, parental cell lines and aneuploid cell lines. These SNPs were required to be heterozygous in the aneuploid line and homozygous in the parental line. Per informative SNP position, we selected all discordant read pairs overlapping the SNP position and indicating the correct rearrangement from the WGS dataset of the aneuploid cell line using sambamba (*Tarasov et al., 2015*). In a separate file we selected all normal read pairs. Subsequently, we counted the reads that correspond to each allele of the respective SNP among the discordant and concordant read pairs. We identified a total of nine phase-informative SNPs, all indicating that the rearrangements affected the transferred chromosome.

## Statistics

We used permutation testing to find out whether unique (*de novo*) CNAs were significantly enriched on the aneuploidy chromosome *versus* the rest of the genome. Therefore, we took the total set of 60 *de novo* CNAs identified in the aneuploid cells by SNP arrays. For each of these 60 *de novo* CNAs we performed a random sampling from the genome. During the random sampling, chromosome size as well as the presence of an extra chromosome (3, 5 or 8) in each of the lines with *de novo* CNAs was considered. After the random sampling, the number of CNAs on the aneuploid chromosome or the remaining chromosomes was calculated. We repeated the above random permutation 1 million times and inferred an empirical p-value from this. A p-value for the difference in the distribution of sizes between *de novo* CNAs on the aneuploidy chromosome versus the rest of the genome was calculated using a Wilcoxon test. To test whether breakpoint orientations observed for the Hte5-1 and Htr8-05 cell lines were deviating from a random distribution (25% for each orientation: tail-head, head-tail, tail-tail, head-head), we used a multinomial goodness of fit test (chisq. test in R). The p-values for the individual cell lines did not meet the customary cut-off of 0.05 when testing the null hypothesis that the number of observations for each orientation is deviating from a random distribution (p=0.1 for Hte5-1; p=0.45 for Htr8-05). However, when combining both datasets, we observed a significant difference at $\alpha = 0.05$ (p=0.03). All other statistical analyses were performed using GraphPad Prism 5.04 (GraphPad Software, San Diego, CA; *p<0.05, **p<0.005, ***p<0.0005, ****p<0.00005.

## Availability of supporting data

The SNP array data set supporting the results of this article is available in the Gene Expression Omnibus under the accession number GSE71979, http://www.ncbi.nlm.nih.gov/geo/query/acc.cgi?acc=GSE71979.

The WGS data set supporting the results of this article is available in the European Nucleotide Archive repository under the accession number PRJEB10264, http://www.ebi.ac.uk/ena/data/view/PRJEB10264.

## Acknowledgements

We thank Edwin Cuppen for help with arranging the genome sequencing, Gaby M de Vries-Simons and Sara Pulit for help with the statistical analysis, Karen Duran for help with isolation of DNA from some of the cell lines and Claudia Herkt, Rosa Ghaisariee, Aaron Gill and other members of the Storchova group for help with the microscopy and image analysis. Hte5-01 was kindly provided by Minoru Koi, University of Michigan, Ann Arbor, Michigan, USA. The work of KK was supported by grant from German Research Foundation to ZS STO 918/7–1 FOR2800. The authors declare no competing interests.

## Additional information

### Funding

| Funder | Grant reference number | Author |
| --- | --- | --- |
| Deutsche Forschungsgemeinschaft | Sto 918 - 5/1 | Markus J van Roosmalen |
| Deutsche Forschungsgemeinschaft | ZS STO 918/7–1 FOR2800 | Kristina Keuper Zuzana Storchova |

The funders had no role in study design, data collection and interpretation, or the decision to submit the work for publication.

### Author contributions

Maja Kneissig, Formal analysis, Supervision, Validation, Investigation, Visualization, Methodology; Kristina Keuper, Data curation, Software, Formal analysis, Investigation, Visualization; Mirjam S de Pagter, Conceptualization, Data curation, Software, Investigation, Visualization, Methodology; Markus J van Roosmalen, Conceptualization, Data curation, Software, Formal analysis, Investigation; Jana Martin, Hannah Otto, Validation, Investigation, Visualization; Verena Passerini, Conceptualization, Supervision, Validation, Investigation; Aline Campos Sparr, Validation, Investigation, Methodology; Ivo Renkens, Software, Formal analysis, Methodology; Fenna Kropveld, Data curation, Software, Formal analysis, Validation, Methodology; Anand Vasudevan, Data curation, Software, Formal analysis, Investigation; Jason M Sheltzer, Software, Supervision, Validation, Investigation; Wigard P Kloosterman, Conceptualization, Data curation, Software, Formal analysis, Funding acquisition, Validation, Investigation; Zuzana Storchova, Conceptualization, Resources, Data curation, Formal analysis, Supervision, Funding acquisition, Validation, Investigation, Methodology, Project administration

### Author ORCIDs

Jason M Sheltzer (iD) http://orcid.org/0000-0003-1381-1323
Zuzana Storchova (iD) https://orcid.org/0000-0003-2376-7047

### Decision letter and Author response

Decision letter https://doi.org/10.7554/eLife.50292.sa1
Author response https://doi.org/10.7554/eLife.50292.sa2

## Additional files

### Supplementary files

• Supplementary file 1. Cell lines used in the study. The cell lines were created by microcell-mediated chromosome transfer. The % of cells with whole chromosome aneuploidy (WCA) shows the % of metaphase spreads that scored positive for the expected WCA.

• Supplementary file 2. List of all identified copy number aberrations.

• Supplementary file 3. The number and percentage of informative SNPs per chromosomal region used for determination of the chromosome affected by unique CNAs; regions are determined per clone, by copy number.

• Supplementary file 4. Breakpoint junctions detected by WGS that were selected for validation by PCR, Sanger sequencing and *de novo* assembly.

• Supplementary file 5. Overview of informative SNPs deduced from whole genome sequencing reads, allowing determination of the affected chromosome.

• Transparent reporting form

### Data availability

High throughput data are available in public repositories. The SNP array data set supporting the results of this article is available in the Gene Expression Omnibus under the accession number GSE71979; the WGS data set supporting the results of this article is available in the European Nucleotide Archive repository under the accession number PRJEB10264. All data generated or analysed during this study are included in the manuscript and supporting files. Source data files have been provided for Figures 1 (in the supplementary files), 2, 3, 4 and 5.

The following datasets were generated:

| Author(s) | Year | Dataset title | Dataset URL | Database and Identifier |
|---|---|---|---|---|
| Kloosterman W | 2015 | Induced genomic rearrangements and chromothripsis by micronucleus-mediated chromosome transfer | https://www.ncbi.nlm.nih.gov/geo/query/acc.cgi?acc=GSE71979 | NCBI Gene Expression Omnibus, GSE71979 |
| Kloosterman W | 2019 | Whole genome sequencing of cell lines generated with micronucleus mediated chromosome transfer | https://www.ebi.ac.uk/ena/data/search?query=PRJEB10264 | European Nucleotide Archive, PRJEB10264 |

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
