## [Decision Letter]

**Acceptance summary:**

Cancer cells often show drastic genome rearrangements. Such rearrangements may arise by 'chromothripsis', where a chromosome is shattered and then pieced together in abnormal ways. Chromothripsis can be triggered when a lagging chromosome gets incorporated into a micronucleus, which does not provide the same functionality as the main nucleus leading to damage of that chromosome. In this paper, Kneissig et al. show that chromothripsis-like events are also observed in microcell-mediated chromosome transfer (MMCT). MMCT introduces one additional copy of any desired human chromosome into human acceptor cells. MMCT has an intermediate stage where the chromosome to be transferred is encapsulated in a micronucleus, and it is presumably during this stage that chromothripsis is initiated. Excitingly, this system therefore provides the opportunity to trigger chromosomal rearrangements of any one specific chromosome. Importantly, MMCT does not always result in chromothripsis. So, some transfer events will introduce an intact chromosome, others will introduce a rearranged chromosome, and the resulting cell lines can be directly compared. Normally an extra chromosome impairs growth of the target cells, but the authors show that target cells tend to grow more readily when that chromosome has been rearranged. Hence, this is an attractive system to study complex genetic defects as they are found in cancer cells and their functional consequences. We think this will be a highly useful tool for researchers studying chromosome instability and cancer.

**Decision letter after peer review:**

Thank you for submitting your article "Micronuclei-based model system reveals functional consequences of chromothripsis in human cells" for consideration by *eLife*. Your article has been reviewed by three peer reviewers, one of whom is a member of our Board of Reviewing Editors, and the evaluation has been overseen by Detlef Weigel as the Senior Editor. The following individuals involved in review of your submission have agreed to reveal their identity: Sarah McClelland (Reviewer #2).

The reviewers have discussed the reviews with one another and the Reviewing Editor has drafted this decision to help you prepare a revised submission.

Summary:

You describe in this manuscript the use of microcell-mediated chromosome transfer (MMCT) to study the consequences of micronuclei formation and the effect of subsequent chromosomal rearrangements on cancer cells. You found that MMCT can induce complex rearrangements of the transferred chromosome that have hallmarks of chromothripsis. Similar rearrangements are observed in cancer cells and can be a consequence of chromosome missegregation-induced micronuclei formation. The cell lines in which rearrangements have occurred in your system can be directly compared to cell lines in which the transferred chromosome has remained intact, which allows you to specifically study the consequences of rearrangements.

All reviewers found this interesting and appreciated the advance that this system provides. Your data suggest that MMCT can be used to create and study the consequences of rearrangements for any chromosome of interest. This is an advance over previous systems, where either only the Y chromosome (small and gene-poor) could be targeted (Ly, Cleveland et al., Nat Cell Biol 2017, Nat Genetics 2019) or where mitotic missegregation is used, which affects different chromosomes largely randomly.

The reviewers felt that the experiments in the first part of your manuscript, describing the chromosomal rearrangements after MMCT, are strong and that your manuscript is very well written. However, some major claims in the second part are not yet well substantiated, as outlined below. To the extent that this is possible, these conclusions need to be backed up with new experimental evidence or additional analyses of existing data.

Essential revisions:

1) You made the interesting observation that smaller micronuclei tend to lack lamin B1. You speculate that this is a consequence of higher curvature in micronuclei with less DNA and that this increases susceptibility to NE rupture. The reviewers felt that it was not sufficiently clear whether the loss of lamin B1 is indeed a direct consequence of less DNA and a higher curvature, or whether the observed correlations (lower DAPI intensity/smaller diameter in lamin B1-negative MN) could result from chromatin loss or compaction after NE rupture has occurred.

We suggest you combine CLS-GFP with lamin B1 and DAPI staining in A9 cells to show that – at least initially – the NE on the lamin B1-negative micronuclei is intact and that lower DNA content/diameter already correlates with less lamin B1 at this early stage. In addition, it would be useful to know if lamin B1-negative MN in tendency contain smaller or fewer chromosomes, which could be done by centromere staining and/or FISH. Furthermore, if the hypothesis is correct, smaller chromosomes may have a higher likelihood for DNA damage and chromothripsis. Can you provide any evidence that supports this?

2) Other questions arose with respect to the time point and reason for DNA damage in the micronuclei in your system:

After colchicine + cytochalasin B treatment of A9 cells, the NE of micronuclei seems to remain intact and DNA damage remains low, although these cells contain lamin B1-negative micronuclei (Figure 3D/E). The only difference between this step of the procedure and the isolated micronuclei, which show NE rupture and more DNA damage (Figure 2J/K, 3), is a centrifugation step. How long does this centrifugation take? And do you think that the mechanical force of centrifugation plays a role in the NE rupture? If so, this seems worthwhile mentioning. Length of the centrifugation matters because cells remain in cytochalasin B at this step. Since cytochalasin B by itself has been reported to fragment DNA (doi: 10.1096/fasebj.4.12.2394319), it would be important to extend the cytochalasin B treatment of A9 cells by this time period for a direct side-by-side comparison between A9 and MN.

In the Discussion "We propose that similar mechanisms act on isolated MN after fusion…" does only seem partially correct. Much of the DNA damage seems to have occurred before fusion.

Related to the NE rupture in MN, Figure 3C should be improved. In the pictures provided, it is unclear where the border of the microcell is and whether nucleo-cytoplasmic compartmentalization can be observed at all (no example is shown). The text mentions that NLS-RFP signals were lost in 84% of micronuclei. Why is that? It would be useful to show examples of all types of patterns observed and include their frequency into the main figure. In addition, it would be helpful to correlate the loss of compartmentalization at this stage with lamin B1 staining and/or DNA damage.

3) Your data interestingly suggest that chromothripsis increases colony formation and you find a positive correlation between reduced DNA amount and colony growth. Please use the data that you already have to more globally report how much DNA is lost upon chromothripsis. Is this a general phenomenon across your many clones? In order to strengthen your conclusion, it would be helpful to increase the number of cell lines analyzed for their colony formation ability. Could the Htr8 lines from Figure 1 be included? In addition, please provide a p value for Figure 4D using slope = 0 as null hypothesis.

4) In Figure 4C, you analyze whether chromothripsis may impact colony formation through affecting the fidelity of mitosis. Your data suggest that cell lines that form more colonies might have fewer anaphase bridges, such as HTr5-16/19 compared with 17 and 18. Analysis of additional cell lines should be possible (at least all those used in 4b/d) and would make the result more conclusive. After analysis of additional cell lines, these data could be plotted against relative colony formation similar to Figure 4D.

Have you considered CNAs on disomic chromosome as another factor that may affect colony formation?

5) You tested the idea that cells with chromothripsis lose genes that reduce colony formation and found COL4A1/2 as one possible example. To strengthen this conclusion, both genes should be expressed in the clone with rearrangement to test the effect on colony formation. Otherwise, there is no evidence that this had any functional consequence.

We expect that you will be able to deepen your analysis in at least three of these areas. For those, where you are unable to do so, you need to soften your claims.

---

## [Author Response]

Essential revisions:1) You made the interesting observation that smaller micronuclei tend to lack lamin B1. You speculate that this is a consequence of higher curvature in micronuclei with less DNA and that this increases susceptibility to NE rupture. The reviewers felt that it was not sufficiently clear whether the loss of lamin B1 is indeed a direct consequence of less DNA and a higher curvature, or whether the observed correlations (lower DAPI intensity/smaller diameter in lamin B1-negative MN) could result from chromatin loss or compaction after NE rupture has occurred.We suggest you combine CLS-GFP with lamin B1 and DAPI staining in A9 cells to show that – at least initially – the NE on the lamin B1-negative micronuclei is intact and that lower DNA content/diameter already correlates with less lamin B1 at this early stage.

We thank the reviewers for this interesting suggestion. In our new experiments, we combined lamin B staining with the CLS-GFP staining and diameter measurements simultaneous in treated A9 cells. This experiment clearly showed that the micronuclei lacking lamin B have a smaller diameter and more often contain CLS-GFP, thus supporting the notion of increased susceptibility to NE rupture. Add the same time, the MNs that lost lamin B1 are consistently smaller than those that contain lamin B1, suggesting that the presence of lamin B1 also affects the size of the MNs. This new data are in a new Figure 4 D-F and in a Figure 2—figure supplement 1E and the corresponding text was altered in the manuscript.

In addition, it would be useful to know if lamin B1-negative MN in tendency contain smaller or fewer chromosomes, which could be done by centromere staining and/or FISH.

To this end, we used an anti-centromeric antibody to label the centromeres in combination with lamin B1 staining and analyzed the correlation between MN size and number of chromosomes in the MN. This experiments revealed that there is a correlation between the MN size and number of chromosomes in the MN. However, we also observed a relationship between lamin B presence and MN size, as all MNs containing lamin B in the NE were consistently larger then MNs with the same number of centromeric markers, but lacking lamin B. This suggests that the lack of lamin B in the envelope prompts a “shrinkage” of the envelope and therefore smaller diameter of the micronucleus. This new results are presented in the Figure 2—figure supplement 1C and the corresponding text was altered accordingly.

It should be noted that the relationship between chromosome size and MN size might be obscured by the differences in the chromosome size (from 50 to 200 Mb). Unfortunately, we were not able to establish the IF-FISH technique that would allow to determine and compare the correlation of lamin B and the MN diameter in the MNs containing the same chromosome.

Furthermore, if the hypothesis is correct, smaller chromosomes may have a higher likelihood for DNA damage and chromothripsis. Can you provide any evidence that supports this?

There is no evidence that smaller chromosome may have a higher likelihood for DNA damage and chromothripsis based on our data, as the sample size is too small for statistical analysis. This is true for all papers on chromothripsis published so far, which is because of the technical difficulties and high expenses associated with analysis of chromothripsis. Analysis of chromothripsis from cancer (e.g. Pallis et al., Oncotarget 2018, Cortés-Ciriano et al., BioRxiv 2018) does not show an apparent bias toward smaller chromosomes, but rather a chromosome specific and cancer specific biases – e.g. chromosome 11 is frequently rearranged in breast cancer. This is likely because of the important role of selection in cancer.

2) Other questions arose with respect to the time point and reason for DNA damage in the micronuclei in your system:After colchicine + cytochalasin B treatment of A9 cells, the NE of micronuclei seems to remain intact and DNA damage remains low, although these cells contain lamin B1-negative micronuclei (Figure 3D/E). The only difference between this step of the procedure and the isolated micronuclei, which show NE rupture and more DNA damage (Figure 2J/K, 3), is a centrifugation step. How long does this centrifugation take? And do you think that the mechanical force of centrifugation plays a role in the NE rupture? If so, this seems worthwhile mentioning. Length of the centrifugation matters because cells remain in cytochalasin B at this step. Since cytochalasin B by itself has been reported to fragment DNA (doi: 10.1096/fasebj.4.12.2394319), it would be important to extend the cytochalasin B treatment of A9 cells by this time period for a direct side-by-side comparison between A9 and MN.

To test directly whether the length of cytocholasin B treatment and centrifugation forces affect the integrity and DNA damage in micronuclei, we performed the experiments as suggested – after disrupting the cells we centrifuged them to isolate the micronuclei. In parallel, we kept the same samples for exactly the same period of time in cytochalasin B without centrifugation and time points were taken similarly. Using γ-H2A.X labeling as a proxy for DNA damage, we found that both centrifugation and the time in cytochalasin B contribute to the accumulation of DNA damage. These new data are now in the Figure 2—figure supplement 1F were incorporated in the text. It should be noted that these new experiments suggested that there is a minor portion of micronuclei that already lost their integrity in A9 cells. We have now added these data to the manuscript and corrected our previous statements that all micronuclei are intact despite lacking lamin B1.

In the Discussion "We propose that similar mechanisms act on isolated MN after fusion…" does only seem partially correct. Much of the DNA damage seems to have occurred before fusion.

We apologize for the imprecise statement. Indeed, most of the DNA damage seems to occur before fusion, however, it should be noted that the quantification of DNA damage after the fusion was not performed because it is technically impossible. We also consider important that the pattern of rearrangements of some of the analyzed chromothripsis events were suggestive of DNA-replication dependent events, which most likely occur after the fusion. We have now corrected the statement as follows:

“Additionally, DNA damage or replication-dependent rearrangements can occur also in MN after fusion to the acceptor cells, as suggested by the finding of patterns of aberrant replicative processes on the rearranged chromosomes”.

Related to the NE rupture in MN, Figure 3C should be improved. In the pictures provided, it is unclear where the border of the microcell is and whether nucleo-cytoplasmic compartmentalization can be observed at all (no example is shown). The text mentions that NLS-RFP signals were lost in 84% of micronuclei. Why is that? It would be useful to show examples of all types of patterns observed and include their frequency into the main figure.

We would like to emphasize here that these are images of isolated micronuclei that are very small and extremely difficult to image. We have also never observed CLS-GFP alone in isolated micronuclei. This is most likely due to extremely small volume – or even an absence – of the cytoplasm. The statement that NLS-RFP was lost is misleading. In fact, the transfection efficiency of both CLS-GFP and NLS-RFP is rather low (transfection efficiency 20 – 22% ) and therefore most cells did not contain any signal. Accordingly, most isolated MNs did not contain any signal. We apologize for this unfortunate formulation that we now corrected:

“In contrast, we observed colocalization of NLS with CLS in 14% of MNs, while NLS alone was detected only in 2% of the isolated MNs (Figure 3C). 84% of the isolated micronuclei were negative for both CLS and NLS, which corresponds with the transfection efficiency”.

In addition, it would be helpful to correlate the loss of compartmentalization at this stage with lamin B1 staining and/or DNA damage.

We performed the suggested experiment to correlate the loss of compartmentalization with lamin B localization and DNA damage. We followed the suggestion of the reviewer and used only transfection with CLS. These new results confirm our previous notion that the lamin B negative cells frequently lose the compartmentalization and often contain DNA damage. These findings are now summarized in Figure 4D–F.

3) Your data interestingly suggest that chromothripsis increases colony formation and you find a positive correlation between reduced DNA amount and colony growth. Please use the data that you already have to more globally report how much DNA is lost upon chromothripsis. Is this a general phenomenon across your many clones?

We are not sure how to globally report on how much DNA is lost upon chromothripsis, because it is a relative “loss” from a gain of a full chromosome. This differs in every individual cell line and depends on the size of the additional chromosome – if a smaller chromosome was added – e.g. 21 – then less DNA is lost. Therefore, we prefer to present the data as the amount of additionally gained DNA. Here, full trisomy contains more additional DNA than trisomy where the trisomic chromosome was rearranged. The “loss” is observed only in clones that underwent chromothripsis and therefore it cannot be considered a general phenomenon.

In order to strengthen your conclusion, it would be helpful to increase the number of cell lines analyzed for their colony formation ability. Could the Htr8 lines from Figure 1 be included?

We thank the reviewers for this interesting suggestion. Unfortunately, some of the aneuploid cell lines (that tend to be in general more sensitive to culturing conditions) did not recover from freezing and could not be used in additional experiments. Nevertheless, we added four additional clones for colony forming assay. The new results did not change the previous general conclusion that the colony forming ability correlates with the amount of extra DNA and were now added to the Figure 5B,D-F and Figure 5—figure supplement 2A,B.

In addition, please provide a p value for Figure 4D using slope = 0 as null hypothesis.

The reviewers recommend to provide a p value for the plot using slope = 0 as null hypothesis. We are not convinced that this is a good idea for several reasons. Most importantly, this p-value is useful only if the calculated slope for linear regression is a good fit. However, in our case, linear slope is a suboptimal fit; in fact binomial function would be much better fit. This is because additional factors affect the relationship between colony formation capacity and the amount of additional material (e.g. identity of the extra chromosome or the specific rearrangements). To avoid the problems with suboptimal linear fit, we now use Spearman correlation coefficient, which is based on ranking and therefore independent of the specific slope fitting. The p-value here tests the strength of the relationship between the variables (and not the difference from the slope=0). All correlation analysis are now calculated in this way and the Spearman correlation coefficient and the corresponding p-value is provided. The data are plotted in a scatter graph and we determined a trend line to present visually the strength of the correlation.

4) In Figure 4C, you analyze whether chromothripsis may impact colony formation through affecting the fidelity of mitosis. Your data suggest that cell lines that form more colonies might have fewer anaphase bridges, such as HTr5-16/19 compared with 17 and 18. Analysis of additional cell lines should be possible (at least all those used in 4b/d) and would make the result more conclusive. After analysis of additional cell lines, these data could be plotted against relative colony formation similar to Figure 4D.

We thank the reviewers for this very interesting idea. We have reanalyzed our data from this point of view and included the additional clones that were requested by the reviewers in point 3. The analysis revealed that the colony forming capacity negatively correlates with both anaphase bridges and micronuclei occurrence (new Figure 5d, figure 5—figure supplement 5A). This means that elevated genomic instability negatively impacts on colony forming ability. Additionally, however, we observed a strong positive correlation between the anaphase bridges and micronuclei formation and the amount of additional genetic material (new Figure 5F, Figure 5—figure supplement 5B). This fits with the previous notion that the strength of the phenotypes of trisomic cells loosely correlates with the amount of extra material (previous data from Amon, Storchova and other laboratories).

Have you considered CNAs on disomic chromosome as another factor that may affect colony formation?

We have not tested the effect of CNAs on disomic chromosomes. While we agree with the reviewer that this might be an important factor, we were not able to design a strategy that would allow us to test the effect of other CNAs.

5) You tested the idea that cells with chromothripsis lose genes that reduce colony formation and found COL4A1/2 as one possible example. To strengthen this conclusion, both genes should be expressed in the clone with rearrangement to test the effect on colony formation. Otherwise, there is no evidence that this had any functional consequence.

We accept the criticism of the reviewers. Unfortunately, we were not able to obtain expression vectors with COL4A1/2. Therefore, we could not perform the experiments that would allow us to directly link this specific rearrangement with the observed proliferation changes. To soften the claims, we have removed this part from the Results and included it in a discussion where it is clearly stated as an untested hypothesis:

“However, effect of specific rearrangements cannot be excluded. For example, individual assessment of the analyzed cell lines revealed that Htr13-03 (with rearranged chromosomes 13) lost the extra copies of COL4A2 and COL4A1, while Htr13-02 (with intact chromosome 13) carries three copies of each (Figure 5—figure supplement 1B). COL4A2 and COL4A1 encode type IV collagen alpha proteins. Upon proteolytic cleavage of COL4A2 and COL4A1, biologically active peptides called Arresten and Canstatin, respectively, are formed that inhibit angiogenesis, proliferation and tumor formation (Egeblad, Rasch, and Weaver, 2010; Kamphaus et al., 2000). Further experiments will be necessary to determine whether specific copy number changes or rearrangements of individual genes could be selected to improve proliferation of aneuploid cells.”

We expect that you will be able to deepen your analysis in at least three of these areas. For those, where you are unable to do so, you need to soften your claims.

As specified above, we have extended our analysis in four of the five above specified areas. Regarding the point 5, we have softened our conclusions. We believe that our additional data sufficiently address the reviewers’ suggestions.